# NF-κB inhibition in keratinocytes causes RIPK1-mediated necroptosis and skin inflammation

Snehlata Kumari[1],*, Trieu-My Van[1],*, Daniela Preukschat[1], Hannah Schuenke[1], Marijana Basic[2], André Bleich[2], Ulf Klein[3], Manolis Pasparakis[1]

Tumor necrosis factor receptor 1 (TNFR1) activates NF-κB–dependent pro-inflammatory gene expression, but also induces cell death by triggering apoptosis and necroptosis. Inhibition of inhibitor of NF-κB kinase (IKK)/NF-κB signaling in keratinocytes paradoxically unleashed spontaneous TNFR1-mediated skin inflammation in mice, but the underlying mechanisms remain poorly understood. Here, we show that TNFR1 causes skin inflammation in mice with epidermis-specific knockout of IKK2 by inducing receptor interacting protein kinase 1 (RIPK1)–dependent necroptosis, and to a lesser extent also apoptosis, of keratinocytes. Combined epidermis-specific ablation of the NF-κB subunits RelA and c-Rel also caused skin inflammation by inducing TNFR1-mediated keratinocyte necroptosis. Contrary to the currently established model that inhibition of NF-κB–dependent gene transcription causes RIPK1-independent cell death, keratinocyte necroptosis, and skin inflammation in mice with epidermis-specific RelA and c-Rel deficiency also depended on RIPK1 kinase activity. These results advance our understanding of the mechanisms regulating TNFR1-induced cell death and identify RIPK1-mediated necroptosis as a potent driver of skin inflammation.

## Introduction

The skin constitutes an essential structural and immunological barrier protecting the organism from microbial challenges. The maintenance of a healthy skin homeostasis requires a proper balance between cell proliferation, differentiation, death, and a tightly regulated cross talk between epithelial and immune cells. Deregulation of this balance results in chronic inflammatory skin diseases, such as psoriasis. Whereas early studies of the mechanisms driving the pathogenesis of inflammatory skin diseases focused on the role of immune cells, recent studies in genetic mouse models have highlighted the importance of keratinocyte-intrinsic mechanisms in regulating skin immune homeostasis and inflammation (Pasparakis et al, 2014). TNF is a potent inducer of inflammation that has emerged as an important therapeutic target in chronic inflammatory diseases, including rheumatoid arthritis, inflammatory bowel disease, and psoriasis (Chaudhari et al, 2001; Peyrin-Biroulet, 2010; Monaco et al, 2015). However, despite the proven clinical efficacy of anti-TNF therapy, the molecular mechanisms by which TNF triggers inflammation in these patients remain poorly understood.

TNF binding to tumor necrosis factor receptor 1 (TNFR1) initiates the formation of a receptor-proximal signaling complex (termed complex I) composed of receptor interacting protein kinase 1 (RIPK1), TNFR1-associated death domain protein (TRADD), TNFR-associated factor 2 (TRAF2), and the E3 ubiquitin ligases cellular inhibitors of apoptosis (cIAP) 1 and 2. Ubiquitination of RIPK1 with K63-linked ubiquitin chains by cIAPs promotes the recruitment of the linear ubiquitin assembly complex (LUBAC), consisting of SHANK-associated RH domain-interacting protein (Sharpin), Heme-oxidized iron-regulatory protein 2 ubiquitin ligase-1 (HOIL-1), and HOIL-1–interacting protein (HOIP), as well as of the transforming growth factor-β-activated kinase 1 (TAK1)/TAK1-binding protein (TAB)1/2 complex. LUBAC ubiquitinates RIPK1 but also other proteins within the complex with linear Ub chains, which allows the recruitment of the IκB kinase (IKK) complex, consisting of the regulatory subunit NF-κB essential modulator (NEMO) and the catalytic subunits IKK2/IKKβ and IKK1/IKKα (Wertz et al, 2015; Ting & Bertrand, 2016; Varfolomeev & Vucic, 2018). IKK is activated within this complex and mediates phosphorylation and subsequent degradation of inhibitor of NF-κB (IκB) proteins resulting in the activation of the transcription factor NF-κB, which is composed of five subunits: RelA, c-Rel, RelB, p50, and p52 that form hetero- and homo-dimers (Oeckinghaus & Ghosh, 2009). NF-κB activation induces the expression of genes promoting immune responses and inflammation but also of genes promoting cell survival and

[1]Institute for Genetics, Cologne Excellence Cluster on Cellular Stress Responses in Aging-Associated Diseases (CECAD) and Center for Molecular Medicine (CMMC), University of Cologne, Cologne, Germany  [2]Institute for Laboratory Animal Science, Hannover Medical School, Hannover, Germany  [3]Division of Haematology and Immunology, Leeds Institute of Medical Research at St. James's, University of Leeds, Leeds, UK

Correspondence: pasparakis@uni-koeln.de; s.kumari@uq.edu.au
Snehlata Kumari's present address is The University of Queensland Diamantina Institute, Translational Research Institute, Brisbane, Australia
*Snehlata Kumari and Trieu-My Van contributed equally to this work

preventing cell death. Indeed, inhibition of NF-κB or destabilization of complex I sensitize cells to TNF-induced death by promoting the formation of death-inducing signaling complexes (Varfolomeev & Vucic, 2018). TNFR1 has been shown to induce caspase-8–dependent apoptosis via two distinct cell death-inducing signaling complexes that depend on either TRADD (complex IIa) or RIPK1 (complex IIb) (Micheau & Tschopp, 2003; Wang et al, 2008; Varfolomeev & Vucic, 2018). When caspase-8 is inhibited, TNFR1 induces necroptosis, an inflammatory type of necrotic cell death, via the RIPK1-dependent activation of RIPK3, which subsequently phosphorylates and activates the pseudokinase mixed lineage kinase like (MLKL) that executes necroptosis by damaging the plasma membrane (Sun et al, 2012; Zhao et al, 2012; Murphy et al, 2013; Cai et al, 2014; Chen et al, 2014; Dondelinger et al, 2014; Pasparakis & Vandenabeele, 2015; Grootjans et al, 2017).

Two distinct checkpoints have been proposed to regulate TNF-induced cell death by acting on distinct cell death-inducing protein complexes. Checkpoint 1 acts at the level of RIPK1, by preventing the activation of its kinase activity via direct phosphorylation by IKK1/2 as well as TBK1, whereas checkpoint 2 prevents TRADD-dependent apoptosis via the NF-κB–dependent expression of pro-survival genes (Justus & Ting, 2015; Ting & Bertrand, 2016; Lafont et al, 2018). According to the currently prevalent model, inhibition of NF-κB–dependent gene transcription triggers TRADD-FADD-caspase-8–dependent apoptosis via complex IIa, whereas inhibition of checkpoint 1 by blocking IKKs or TBK1 or destabilization of complex I by inhibition of cIAPs or LUBAC, triggers RIPK1-FADD-caspase-8–mediated apoptosis via complex IIb (Justus & Ting, 2015; Ting & Bertrand, 2016). Under conditions of caspase-8 inhibition, complex IIb engages RIPK3 to form the necrosome resulting in MLKL-dependent necroptosis (Pasparakis & Vandenabeele, 2015; Grootjans et al, 2017). Consistent with this model, inhibition of RIPK1 kinase activity was shown to prevent skin inflammation in mice lacking sharpin (Sharpin^cpdm/cpdm), by inhibiting TNFR1-mediated FADD-caspase-8–dependent keratinocyte apoptosis (Berger et al, 2014; Kumari et al, 2014; Laurien et al, 2020; Webster et al, 2020).

Mice with epidermis-specific deletion of IKK2 (Ikk2^fl/fl K14-Cre^tg/wt, hereafter referred to as IKK2^E-KO) develop severe psoriasis-like skin inflammation characterized by epidermal hyperplasia, altered keratinocyte differentiation, up-regulation of cytokines and chemokines, and accumulation of immune cells in the skin (Pasparakis et al, 2002; Stratis et al, 2006a, 2006b; Kumari et al, 2013). Keratinocyte-specific ablation of TNFR1 completely prevented inflammatory skin lesion development in IKK2^E-KO mice (Kumari et al, 2013). Furthermore, mice with keratinocyte-specific knockout of both RelA and c-Rel (Rela^fl/fl c-Rel^fl/fl K14-Cre^tg/wt, hereafter referred to as RelA^E-KO c-Rel^E-KO mice) also developed inflammatory skin lesions resembling the phenotype of IKK2^E-KO mice (Grinberg-Bleyer et al, 2015). These results showed that, paradoxically considering the established pro-inflammatory role of NF-κB, inhibition of IKK/NF-κB signaling in the epidermis triggers psoriasis-like skin inflammation in mice. The mechanisms by which IKK/NF-κB signaling in keratinocytes prevents inflammation have remained poorly understood. Here, we provide evidence that inhibition of IKK or NF-κB signaling induced skin inflammation by triggering keratinocyte death primarily by necroptosis. Surprisingly, and contrary to the currently accepted model of two distinct checkpoints regulating TNF-induced cell death, we found that RIPK1 kinase

activity inhibition prevented skin inflammation in both IKK2^E-KO and RelA^E-KO c-Rel^E-KO mice. These in vivo findings identify keratinocyte necroptosis as a key driver of TNF-mediated skin inflammation and provide a paradigm shift in our understanding of the mechanisms by which IKK and NF-κB signaling regulate TNF-induced cell death.

# Results

## TNF induces death in IKK2-deficient keratinocytes

To address whether IKK2 deficiency induces death of epidermal keratinocytes, we stained skin sections from IKK2^E-KO and littermate control mice at postnatal days 1 (P1), P4 and P8 with antibodies against activated cleaved caspase-3 (CC3) and with terminal deoxynucleotidyl transferase dUTP nick end labeling (TUNEL). The skin of IKK2^E-KO mice showed a progressive increase in the number of CC3 and TUNEL-positive cells in the epidermis (Fig 1A), suggesting that IKK2 deficiency triggers death of keratinocytes. To assess whether death of IKK2-deficient keratinocytes is triggered by TNF, we first isolated primary keratinocytes from newborn control and IKK2^E-KO pups and stimulated them with TNF. Indeed, IKK2-deficient keratinocytes showed enhanced TNF-induced cell death compared with control keratinocytes (Fig 1B). To assess the role of TNFR1 in vivo we examined skin sections from mice lacking both IKK2 and TNFR1 in the epidermis (IKK2^E-KO TNFR1^E-KO mice) and found that TNFR1 ablation prevented keratinocyte death (Fig 1C). Therefore, TNFR1 induces keratinocyte death in IKK2^E-KO mice. To gain insight into the potential mechanisms by which IKK2 prevents keratinocyte death, we measured the expression of the pro-survival genes Cflar, Birc2, Birc3, and Birc5 in the epidermis of IKK2^E-KO mice. Cflar and Birc2 mRNA expression was reduced in the epidermis of IKK2^E-KO mice compared with their littermate controls at P4 (Fig 1D), which could contribute to the sensitization of IKK2-deficient keratinocytes to TNF-induced cell death in vivo. In addition, we detected up-regulation of cytokines and chemokines, including Il-1b, Il-6, Il-24, Ccl3, and Ccl4, in the epidermis of IKK2^E-KO mice at P4, indicating ongoing inflammation alongside the death of keratinocytes in IKK2^E-KO mice (Fig 1E).

Microscopically, skin lesion development in IKK2^E-KO mice starts between P3-P4, suggesting that colonization of the skin with commensal bacteria could provide the trigger. To address whether the microbiota could potentially contribute to skin lesion development in IKK2^E-KO mice, we analyzed IKK2^E-KO mice raised under germ-free (GF) conditions. These experiments showed that GF IKK2^E-KO mice developed similar inflammatory skin lesions as seen in IKK2^E-KO mice raised under specific pathogen-free conditions, characterized with increased epidermal thickness, elevated expression of keratin 6 (K6) and K14, reduced expression of K10, and increased number of dying keratinocytes (Fig S1A and B). Therefore, microbiota colonization is not required for the postnatal onset of skin inflammation in IKK2^E-KO mice.

## Inhibition of both apoptosis and necroptosis prevents skin inflammation in IKK2^E-KO mice

TNFR1 induces cell death by activating both FADD-caspase-8–dependent apoptosis and RIPK3-MLKL–dependent necroptosis. To assess whether TNFR1-mediated FADD-caspase-8–dependent

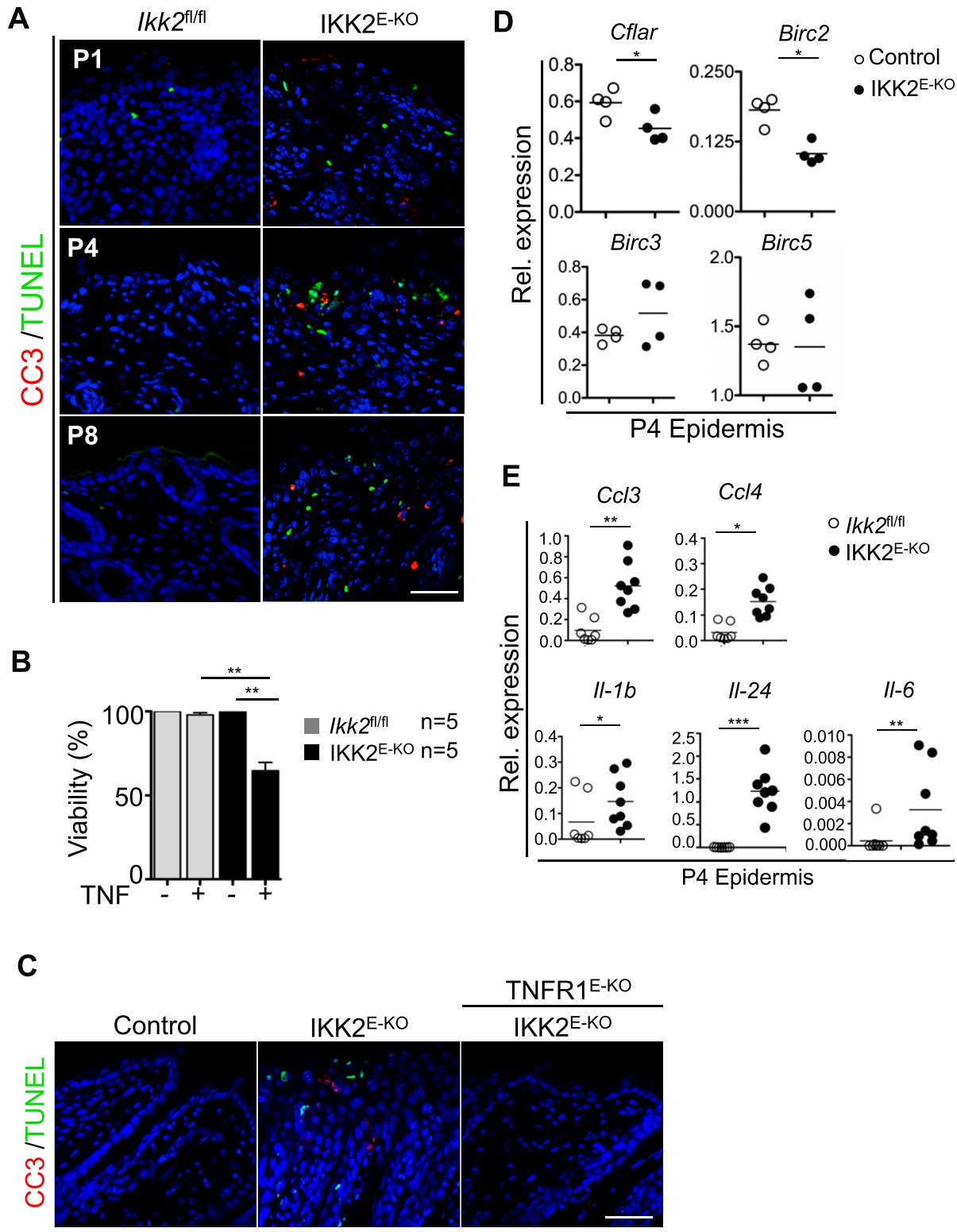

**Figure 1.   IKK2 deficiency induces TNF–mediated death of primary keratinocytes.**

**(A)** Skin sections from P1-P8 pups stained with TUNEL and CC3. Representative images are shown (control and IKK2^E-KO n = 3). Scale bars, 50 μm. **(B)** Primary keratinocytes from Control or IKK2^E-KO (n = 6) mice were treated with TNF (20 ng/ml) for 18 h. Cell viability was determined by WST-1 assay. Graphs show mean ± SEM from pooled data from three independent experiments. Multiple comparisons of groups were evaluated by Kruskal–Wallis one-way ANOVAs with post-Dunn corrections. **(C)** Skin sections from P7-P8 pups stained with TUNEL and CC3. Representative images are shown (control, IKK2^E-KO and IKK2^E-KO TNFR1^E-KO n = 3). Scale bars, 50 μm. **(D, E)** qRT-PCR analysis of the mRNA expression of the indicated genes in RNA isolated from the P4 epidermis of the mice with the indicated genotypes. *$P \leq 0.05$; **$P \leq 0.01$; ***$P \leq 0.005$.

Source data are available for this figure.

apoptosis contributes to skin inflammation we crossed the IKK2$^{E-KO}$ mice with *Fadd*$^{fl/fl}$ animals to generate mice lacking both IKK2 and FADD in keratinocytes. These IKK2$^{E-KO}$ FADD$^{E-KO}$ mice developed severe inflammatory skin lesions at P6-P7 (Fig S2A), showing that FADD deficiency could not prevent skin inflammation in IKK2$^{E-KO}$ mice. Consistently, Z-VAD could not prevent the TNF-induced death in IKK2-deficient primary keratinocytes (Fig S2B). Because epidermis-specific FADD deficiency itself triggers severe skin inflammation by sensitizing keratinocytes to RIPK3-mediated necroptosis (Bonnet et al, 2011), we could not conclude from these results whether FADD deficiency could not prevent skin inflammation in IKK2$^{E-KO}$ mice or if the phenotype of the IKK2$^{E-KO}$ FADD$^{E-KO}$ mice was induced by FADD ablation alone in the epidermis. Therefore, to address the role of keratinocyte death in skin inflammation in IKK2$^{E-KO}$ mice, we generated *Ikk2*$^{fl/fl}$ *K14-Cre*$^{tg/wt}$ *Fadd*$^{fl/fl}$ *Ripk3*$^{-/-}$ mice (hereafter referred to as IKK2$^{E-KO}$ FADD$^{E-KO}$ *Ripk3*$^{-/-}$), which lack IKK2 and FADD specifically in keratinocytes and RIPK3 in all cells. FADD deficiency prevents caspase-8–mediated apoptosis and RIPK3 deficiency prevents necroptosis; therefore, if death of keratinocytes would be important for the pathogenesis of the skin lesions, we would expect that IKK2$^{E-KO}$ FADD$^{E-KO}$ *Ripk3*$^{-/-}$ should not develop the pathology. Indeed, the triple mutant IKK2$^{E-KO}$ FADD$^{E-KO}$ *Ripk3*$^{-/-}$ mice were indistinguishable from their littermate controls and did not show skin lesions at the age of 7–8 d after birth (Fig 2A). Histological analysis of skin sections confirmed that IKK2$^{E-KO}$ FADD$^{E-KO}$ *Ripk3*$^{-/-}$ mice did not show death of keratinocytes and did not develop epidermal hyperplasia and inflammation at P7-P8 (Fig 2A and B). The IKK2$^{E-KO}$ FADD$^{E-KO}$ *Ripk3*$^{-/-}$ mice remained healthy without showing any signs of skin inflammation at least until the age of 1 yr (Fig 2A and Table S1). Furthermore, the epidermis of IKK2$^{E-KO}$ FADD$^{E-KO}$ *Ripk3*$^{-/-}$ mice at P4 did not show up-regulation of cytokine and chemokine expression in contrast to the epidermis of IKK2$^{E-KO}$ mice (Fig 2C). Therefore, combined deficiency in FADD and RIPK3 completely prevented the development of skin lesions in IKK2$^{E-KO}$ mice, demonstrating that death of keratinocytes by apoptosis and/or necroptosis caused skin inflammation in these mice.

### RIPK3-MLKL–dependent keratinocyte necroptosis is the major driver of skin inflammation in IKK2$^{E-KO}$ mice

Having established cell death as an essential trigger for the development of skin lesions in IKK2$^{E-KO}$ mice, we wondered whether the pathology is triggered by keratinocyte apoptosis or necroptosis. Necroptosis is generally considered an inflammatory type of cell death as opposed to apoptosis that is not considered a strong inducer of inflammation. However, previous studies showed that keratinocyte apoptosis drives skin inflammation in *Sharpin*$^{cpdm/cpdm}$, whereas necroptosis plays a minor role in this model (Kumari et al, 2014; Rickard et al, 2014; Webster et al, 2020), suggesting that interfering with complex I function in keratinocytes primarily drives keratinocyte apoptosis.

To assess the specific function of necroptosis, we crossed IKK2$^{E-KO}$ mice with RIPK3-deficient animals. Macroscopic examination revealed a strong amelioration of the skin pathology in IKK2$^{E-KO}$ *Ripk3*$^{-/-}$ mice, which showed only a few small patches of mildly affected skin at P7-P8 as opposed to the severe skin lesions observed in IKK2$^{E-KO}$ mice at this age (Fig 3A). Moreover, histological

analysis of skin sections revealed only small patches showing mildly increased epidermal thickness and inflammation in IKK2$^{E-KO}$ *Ripk3*$^{-/-}$ mice compared with the severely affected skin of IKK2$^{E-KO}$ mice (Fig 3B and C). Immunofluorescence staining of skin sections from IKK2$^{E-KO}$ *Ripk3*$^{-/-}$ mice at P7-P8 showed only mild alterations of epidermal differentiation markers and presence of a few CC3 and TUNEL positive cells in the epidermis, accompanied with mild accumulation of immune cells in the dermis (Fig 3B and C). Furthermore, RIPK3 deficiency strongly prevented the expression of cytokines and chemokines in the epidermis of IKK2$^{E-KO}$ mice at P4, consistent with the histological assessment (Fig 3D). However, IKK2$^{E-KO}$ *Ripk3*$^{-/-}$ mice started to develop progressive skin lesions at the age of about 2–3 mo, which however remained mild compared with the severe skin inflammation of IKK2$^{E-KO}$ mice (Fig 3A and Table S2). These findings suggested that RIPK3-dependent necroptosis plays an important role for skin lesion development in IKK2$^{E-KO}$ mice. However, RIPK3 was reported to regulate inflammation in a cell death–independent manner by acting in myeloid cells (Moriwaki et al, 2014, 2017; Moriwaki & Chan, 2017). We therefore addressed the keratinocyte-intrinsic function of RIPK3 by generating *Ikk2*$^{fl/fl}$ *Ripk3*$^{fl/fl}$ *K14-Cre*$^{tg/wt}$ mice (IKK2$^{E-KO}$ RIPK3$^{E-KO}$). Epidermal keratinocyte-specific RIPK3 knockout fully recapitulated the effect of systemic RIPK3 deficiency, with IKK2$^{E-KO}$ RIPK3$^{E-KO}$ showing strongly ameliorated skin inflammation at P7-P8 but developing progressive mild skin lesions at the age of about 2–5 mo (Fig 3A and B and Table S3). Because RIPK3 has been reported to also mediate necroptosis-independent functions (Moriwaki & Chan, 2017), we further assessed whether RIPK3 deficiency prevented the death of IKK2-deficient keratinocytes. Indeed, RIPK3 deficiency partially inhibited the TNF-induced death of keratinocytes lacking IKK2 (Fig S2C). In addition, the primary keratinocytes from IKK2$^{E-KO}$ FADD$^{E-KO}$ *Ripk3*$^{-/-}$ mice were fully protected from TNF-induced death (Fig S2C), showing that IKK2 deficiency sensitizes keratinocytes to both necroptosis and apoptosis. Taken together, these results showed that RIPK3 acts in a keratinocyte-intrinsic manner to drive skin inflammation in IKK2$^{E-KO}$ mice, suggesting that keratinocyte necroptosis drives the pathology.

To unequivocally address the role of necroptosis in vivo, we crossed the IKK2$^{E-KO}$ mice with mice lacking MLKL, the terminal executioner of necroptosis (Sun et al, 2012). IKK2$^{E-KO}$ *Mlkl*$^{-/-}$ mice were strongly protected from skin inflammation at P7-P8, similarly to IKK2$^{E-KO}$ *Ripk3*$^{-/-}$ and IKK2$^{E-KO}$ RIPK3$^{E-KO}$ mice (Fig 4A). Histological and immunofluorescence analysis of skin sections revealed small patches with slightly increased epidermal thickness, mild alterations in epidermal differentiation markers, as well as the presence of a few dying cells in the epidermis of IKK2$^{E-KO}$ *Mlkl*$^{-/-}$ mice (Fig 4A and B). MLKL deficiency also strongly reduced the expression of cytokines and chemokines expression in the epidermis of IKK2$^{E-KO}$ mice (Fig 4C). Moreover, IKK2$^{E-KO}$ *Mlkl*$^{-/-}$ mice started to develop progressive skin lesions at the age of 2–3 mo similarly to the IKK2$^{E-KO}$ *Ripk3*$^{-/-}$ and IKK2$^{E-KO}$ RIPK3$^{E-KO}$ mice (Fig 4A and Table S4). Collectively, these results revealed that RIPK3-MLKL–dependent necroptosis of keratinocytes constitutes the major trigger of skin inflammation in IKK2$^{E-KO}$ mice particularly early after birth; however, necroptosis-independent mechanisms drive skin lesion development later in life. Our findings that IKK2$^{E-KO}$ FADD$^{E-KO}$ *Ripk3*$^{-/-}$ mice were fully protected from skin lesion development demonstrate

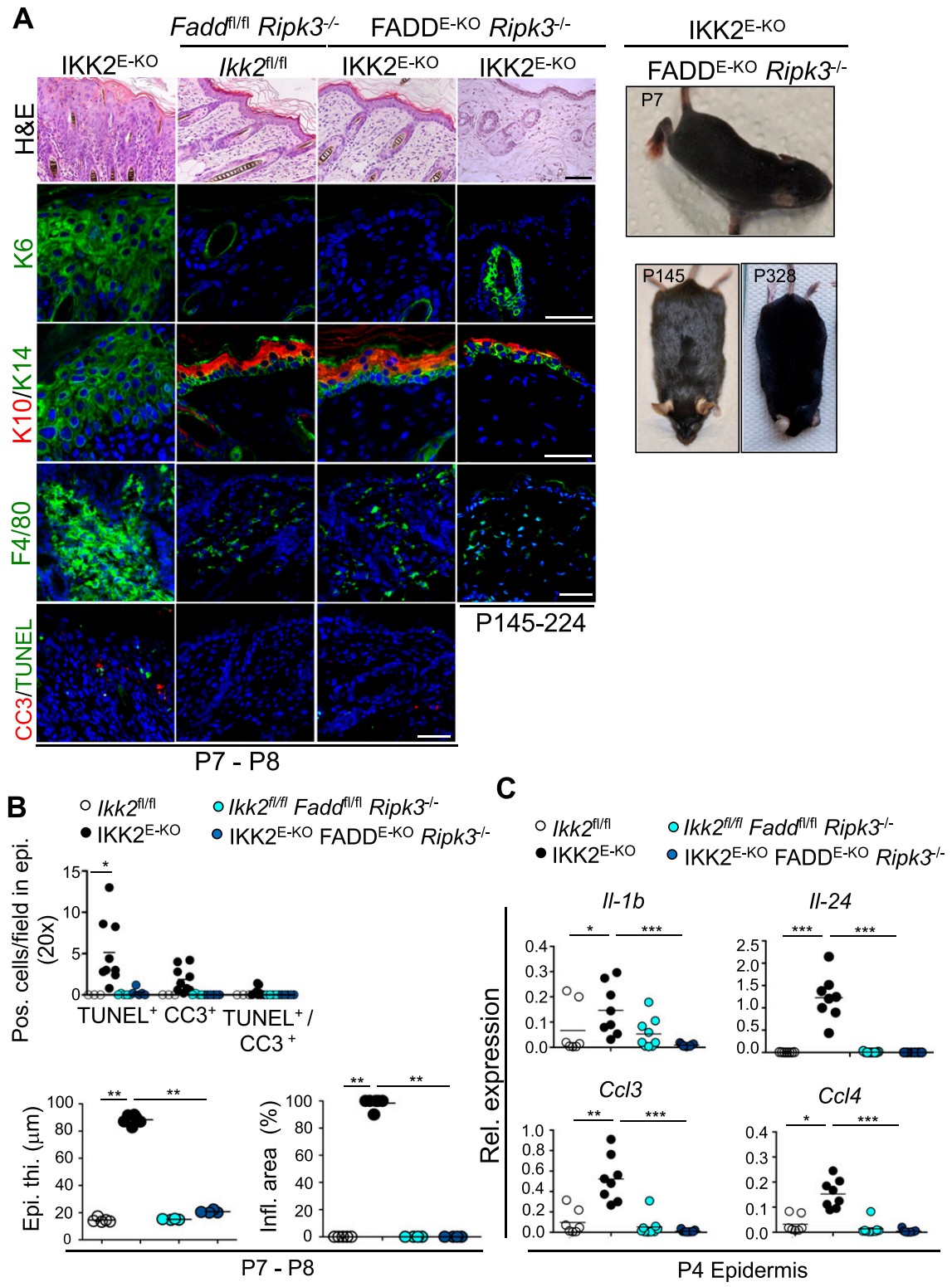

**Figure 2. Inhibition of RIPK3 and epidermal FADD prevents skin inflammation in IKK2[E-KO] mice.**
**(A)** Representative macroscopic mouse pictures and skin sections from mice with the indicated age and genotype stained with H&E or immunostained with the indicated antibodies. Representative images are shown (IKK2[E-KO] n = 6 for H&E, and n ≥ 5 for immunostainings; IKK2[E-KO] FADD[E-KO] Ripk3[-/-] n = 9 [P7-P8] & n = 22 [P134-P385] for H&E, and n ≥ 3 for immunostainings). Scale bars, 50 μm. **(B)** Microscopic quantification of epidermal thickness (Epi. th.) and inflamed skin area (Infl. area) as well as quantification of TUNEL & CC3 positive cells on the skin sections from 7- to 8-d-old mice with the indicated genotypes. **(C)** qRT-PCR analysis of the mRNA expression of the indicated cytokines and chemokines in RNA isolated from the epidermis of 4-d-old mice with the indicated genotypes. Each dot represents individual mice.
Source data are available for this figure.

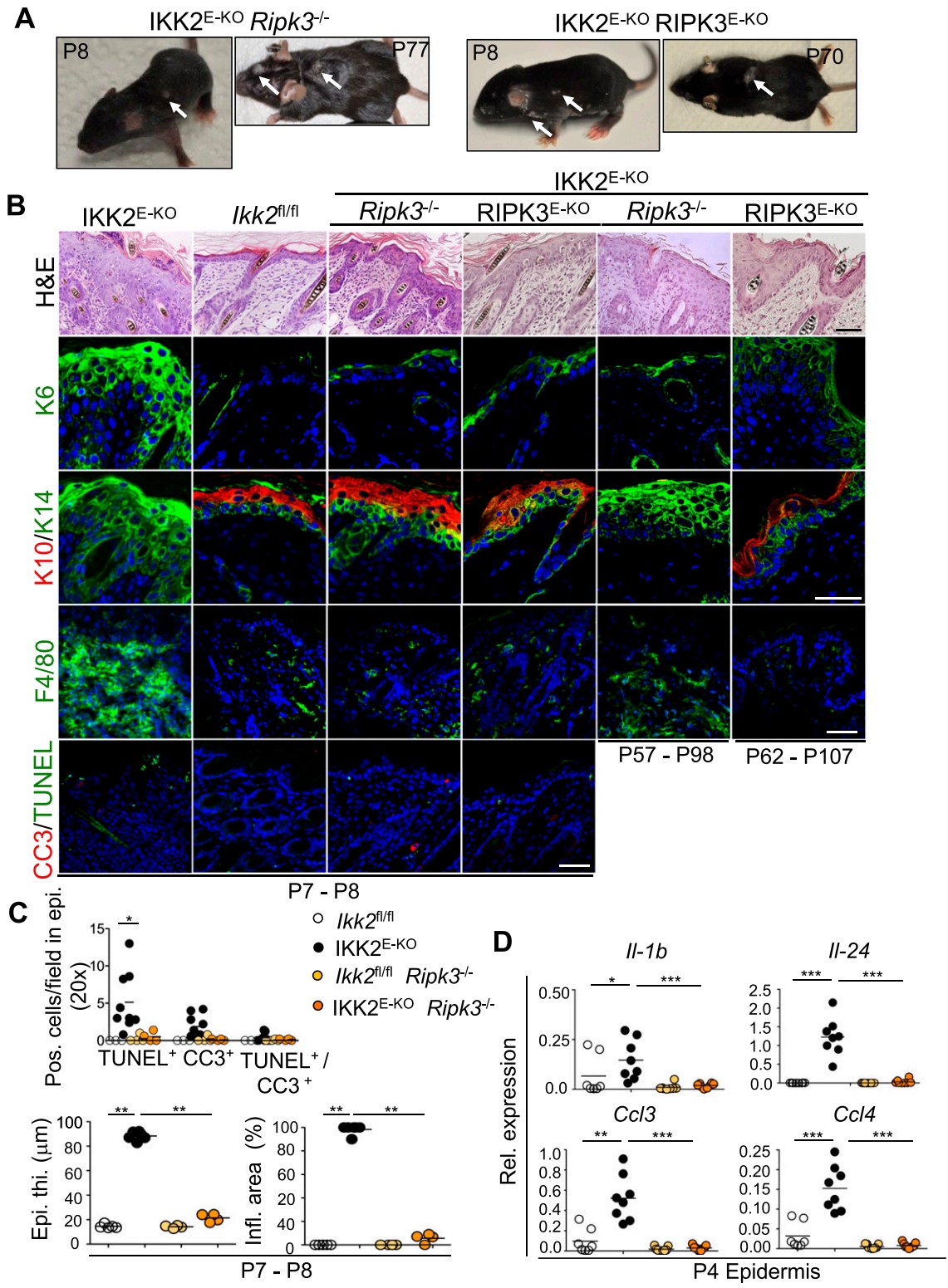

Figure 3. RIPK3-mediated keratinocyte death drives skin inflammation in IKK2^E-KO mice.
(A, B) Representative macroscopic mouse pictures and skin sections from mice with the indicated age and genotype stained with H&E or immunostained with the indicated antibodies. Representative images are shown (IKK2^E-KO n = 6 for H&E, and n ≥ 5 for immunostainings; IKK2^E-KO Ripk3^−/− n = 8 [P7-P8] & n = 16 [P42-P98] for H&E, n ≥ 3 for immunostainings; IKK2^E-KO RIPK3^E-KO n = 7 [P7-P8] & n = 12 [P58-P191] for H&E, and n ≥ 3 for immunostainings; IKK2^E-KO Mlkl^−/− n = 9 [P7-P8] & n = 7 [P66-P103] for H&E, and n ≥ 3 for immunostainings). Scale bars, 50 μm. (C) Microscopic quantification of epidermal thickness (Epi. th.) and inflamed skin area (Infl. area) as well as quantification of TUNEL and CC3-positive cells on the skin sections from 7- to 8-d-old mice with the indicated genotypes. (D) qRT-PCR analysis of the mRNA expression of the indicated cytokines and chemokines in RNA

that FADD-caspase-8–dependent apoptosis also contributes to skin lesion development when necroptosis is blocked.

## RIPK1 kinase activity drives keratinocyte death and skin inflammation in IKK2[E-KO] mice

RIPK1 kinase activity regulates both apoptosis and necroptosis downstream of TNFR1 signaling (Christofferson et al, 2014; Pasparakis & Vandenabeele, 2015). Therefore, to address the role of RIPK1 kinase activity in the pathogenesis of skin lesions in IKK2[E-KO] mice, we crossed them with knock-in mice expressing kinase inactive RIPK1 (*Ripk1*[D138N/D138N]) (Polykratis et al, 2014). IKK2[E-KO] *Ripk1*[D138N/D138N] mice did not show any macroscopically visible skin lesions at P7-P8, when all IKK2[E-KO] mice displayed severely inflamed skin (Fig 5A). In addition, histological and immunofluorescence examination of skin sections from IKK2[E-KO] *Ripk1*[D138N/D138N] mice showed normal epidermal thickness, differentiation marker expression and F4/80[+] immune cell infiltration (Fig 5A and B). Furthermore, lack of RIPK1 kinase activity prevented the up-regulation of cytokine and chemokine expression, including *IL-1b*, *Il-24*, *Ccl3*, and *Ccl4*, in the epidermis of IKK2[E-KO] mice at P4 (Fig 5C). In addition, TUNEL and CC3-positive cells were not detected in the epidermis of IKK2[E-KO] *Ripk1*[D138N/D138N] mice, whereas lack of RIPK1 kinase activity also prevented the TNF-induced death of primary IKK2-deficient keratinocytes (Figs 5A and B and S2C). Most of the IKK2[E-KO] *Ripk1*[D138N/D138N] mice reached adulthood without showing signs of skin inflammation, with only a few of these mice showing mild skin lesions between 5 and 9 mo of age (Fig 5A and Table S5). Taken together, these results showed that RIPK1 kinase activity drives keratinocyte necroptosis and apoptosis and triggers the development of inflammatory skin lesions in IKK2[E-KO] mice.

## NF-κB inhibition causes skin inflammation by inducing RIPK1-dependent keratinocyte necroptosis

IKK2 is the main catalytic subunit of the IKK complex that is responsible for activation of canonical NF-κB by phosphorylating and triggering the degradation of IκBα, resulting in the nuclear translocation of NF-κB dimers and the transcriptional induction of NF-κB target genes (Oeckinghaus & Ghosh, 2009). NF-κB–dependent expression of pro-survival genes has been shown to prevent TNF-induced apoptosis (Beg & Baltimore, 1996; Stehlik et al, 1998; Wang et al, 1998; Barkett & Gilmore, 1999; Chen et al, 2000; Karin & Lin, 2002). However, IKK2 also inhibits RIPK1-mediated apoptosis and necroptosis by directly phosphorylating RIPK1 to suppress its kinase activity (Dondelinger et al, 2015, 2019). Therefore, we wanted to address whether epithelial-specific IKK2 ablation causes keratinocyte death and skin inflammation directly by phosphorylating RIPK1 or by inducing NF-κB–dependent pro-survival gene transcription. To this end, we generated mice with epidermis-specific knockout of RelA and c-Rel, the two NF-κB subunits that are responsible for the transcriptional activation of canonical NF-κB target genes, by crossing *Rela*[fl/fl] *c-Rel*[fl/fl] mice with K14-Cre. These RelA[E-KO] c-Rel[E-KO] mice were indistinguishable from their *Rela*[fl/fl] *c-Rel*[fl/fl] littermate controls at birth but started

to develop inflammatory skin lesions at around P4, which progressed to widespread inflammatory skin pathology by P8-P11. The inflammatory skin lesions of RelA[E-KO] c-Rel[E-KO] mice resembled the phenotype of IKK2[E-KO] mice, although the pathology was generally milder (Fig 6A). Histological and immunofluorescence analysis of skin sections revealed epidermal hyperplasia, altered expression of epidermal differentiation markers and accumulation of immune cells in the skin of RelA[E-KO] c-Rel[E-KO] mice at P8-P11 (Fig 6B and C). Moreover, increased numbers of dying keratinocytes were detected by TUNEL and CC3 staining in the epidermis of RelA[E-KO] c-Rel[E-KO] mice (Fig 6B). Consistently, primary keratinocytes from RelA[E-KO] c-Rel[E-KO] mice showed increased death in response to TNF stimulation (Fig 6D). Therefore, combined ablation of RelA and c-Rel caused keratinocyte death and skin inflammation recapitulating the skin pathology caused by epithelial IKK2 deficiency.

We then wondered if keratinocyte necroptosis drives skin inflammation in RelA[E-KO] c-Rel[E-KO] mice similarly to IKK2[E-KO] mice. Indeed, MLKL deficiency strongly protected RelA[E-KO] c-Rel[E-KO] mice from the development of inflammatory skin lesions. RelA[E-KO] c-Rel[E-KO] *Mlkl*[−/−] mice did not show macroscopically visible skin lesions at P8-P12, in contrast to the severe skin lesions developing in RelA[E-KO] c-Rel[E-KO] mice at this age (Fig 6A). Immunohistological analysis of skin sections from RelA[E-KO] c-Rel[E-KO] *Mlkl*[−/−] mice at P8-P12 revealed a normal skin without signs of hyperplasia and inflammation, confirming that MLKL deficiency prevented skin lesion development in RelA[E-KO] c-Rel[E-KO] mice (Fig 6A–C). RelA[E-KO] c-Rel[E-KO] *Mlkl*[−/−] mice reached adulthood without showing macroscopically visible skin pathology but started to develop skin lesions at about 3–4 mo of age, which remained very mild compared with the skin lesions of RelA[E-KO] c-Rel[E-KO] mice (Fig 6A and B and Table S6). Therefore, MLKL deficiency strongly prevented skin lesion development in RelA[E-KO] c-Rel[E-KO] mice demonstrating that necroptosis is a major driver of the pathology. Of note, the effect of MLKL deficiency in preventing skin lesion development was much stronger in RelA[E-KO] c-Rel[E-KO] mice compared with the IKK2[E-KO] mice.

Previous studies primarily using in vitro cell culture approaches showed that NF-κB or IKK inhibition sensitized cells to TNF-induced death via different mechanisms: NF-κB inhibition caused RIPK1-independent death, whereas inhibition of IKK activity caused RIPK1-dependent cell death (Dondelinger et al, 2015, 2019; Justus & Ting, 2015). Based on these studies, we expected that inhibition of RIPK1 kinase activity should not prevent keratinocyte death and skin inflammation caused by NF-κB inhibition in RelA[E-KO] c-Rel[E-KO] mice. To assess the role of RIPK1 kinase activity, we generated and analyzed RelA[E-KO] c-Rel[E-KO] *Ripk1*[D138N/D138N] mice. Contrary to expectations, inhibition of RIPK1 kinase activity strongly prevented the development of inflammatory skin lesions in RelA[E-KO] c-Rel[E-KO] mice. RelA[E-KO] c-Rel[E-KO] *Ripk1*[D138N/D138N] mice did not show any macroscopically or histologically detected signs of skin inflammation at P8-P12 (Fig 6A–C). RelA[E-KO] c-Rel[E-KO] *Ripk1*[D138N/D138N] mice remained healthy during the first months of life; however, some of these animals developed mild to moderate skin lesions between the age of 6–9 mo (Fig 6A and B and Table S7). To address whether

isolated from the epidermis of 4-d-old mice with the indicated genotypes (data for *Ikk2*[fl/fl] and IKK2[E-KO] mice are same as in Fig 1 and are shown for comparison). Each dot represents individual mice. *P ≤ 0.05; **P ≤ 0.01; ***P ≤ 0.005. Arrows indicate skin lesions. Source data are available for this figure.

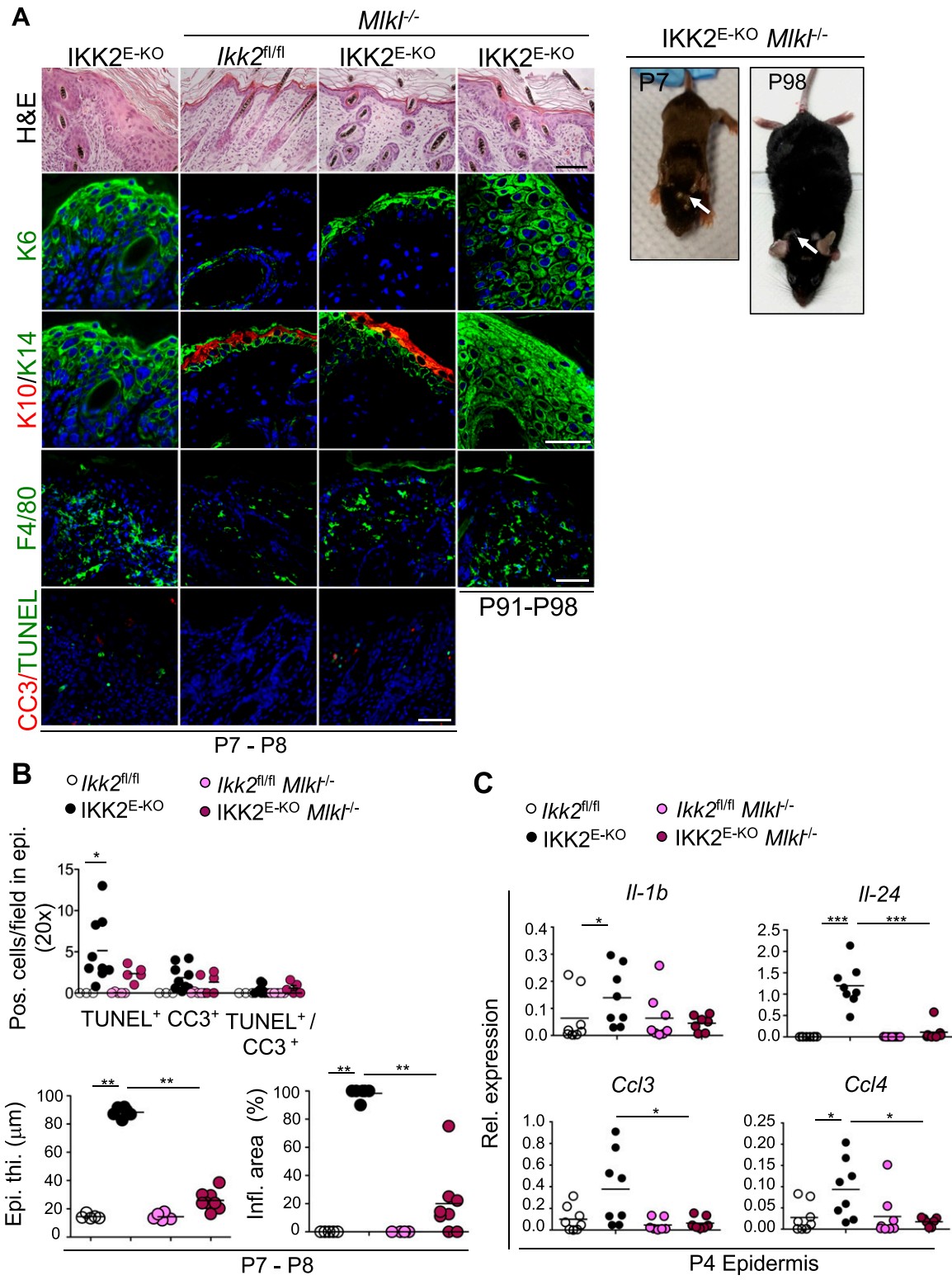

**Figure 4. Mixed lineage kinase like-dependent necroptosis drives skin inflammation in IKK2^E-KO mice.**
**(A)** Representative macroscopic mouse pictures and skin sections from mice with the indicated age and genotype stained with H&E or immunostained with the indicated antibodies. Representative images are shown (IKK2^E-KO n = 6 for H&E, and n ≥ 5 for immunostainings; IKK2^E-KO *Mlkl*^−/− n = 9 [P7-P8] & n = 7 [P66-P103] for H&E, and n ≥ 3 for immunostainings). Scale bars, 50 μm. **(B)** Microscopic quantification of epidermal thickness (Epi. th.) and inflamed skin area (Infl. area) as well as quantification of TUNEL & CC3 positive cells on the skin sections from 7- to 8-d-old mice with the indicated genotypes. **(C)** qRT-PCR analysis of the mRNA expression of the indicated cytokines and chemokines in RNA isolated from the epidermis of 4-d-old mice with the indicated genotypes (data for *Ikk2*^fl/fl and IKK2^E-KO mice are same as in Fig 1 and are shown for comparison). Each dot represents individual mice. *P ≤ 0.05; **P ≤ 0.01; ***P ≤ 0.005. Arrows indicate skin lesions. Source data are available for this figure.

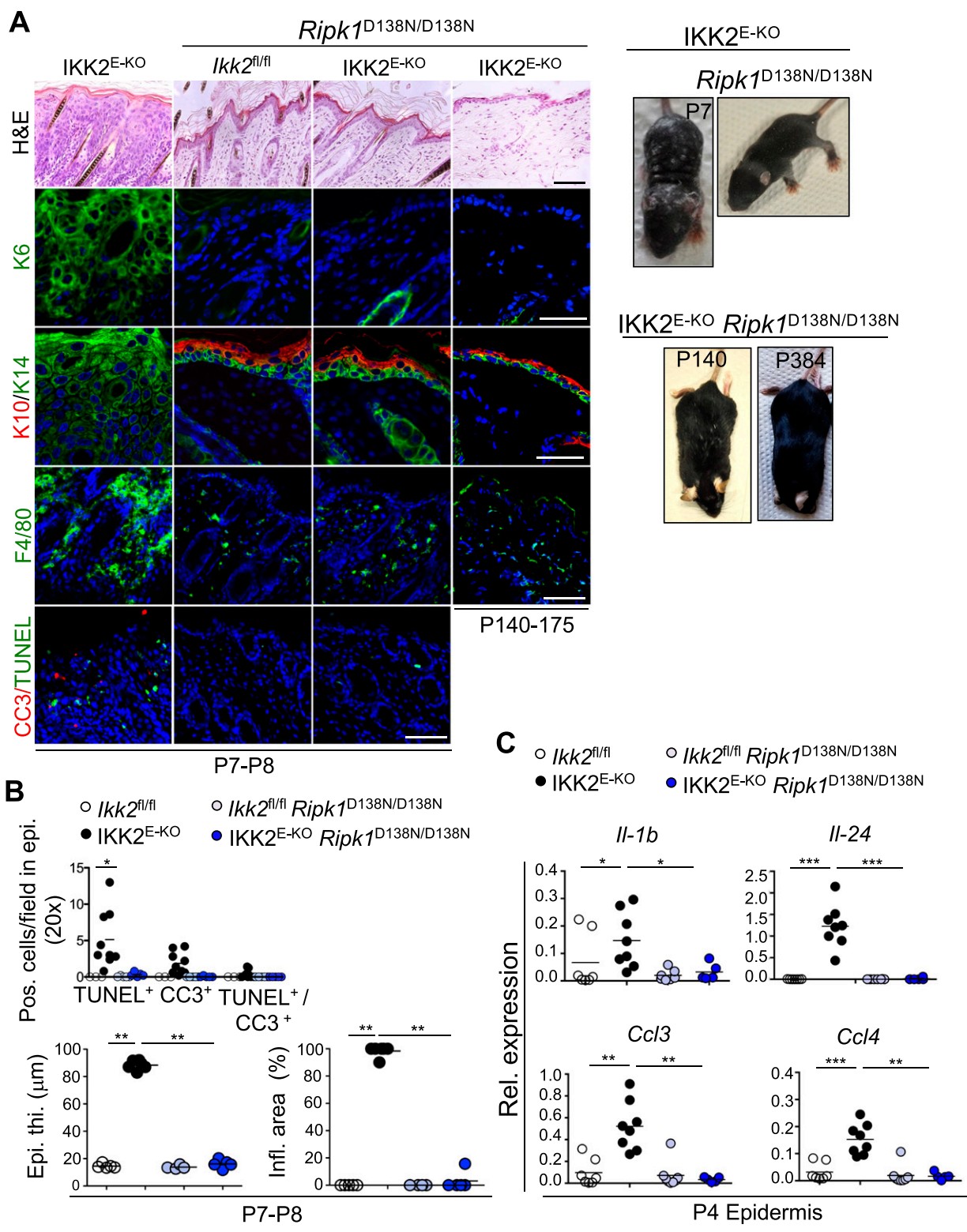

**Figure 5. Kinase activity of RIPK1 triggers skin inflammation in IKK2^E-KO mice.**
**(A)** Representative macroscopic mouse pictures and skin sections from mice with the indicated age and genotype stained with H&E or immunostained with the indicated antibodies. Representative images are shown (IKK2^E-KO n = 6 for H&E and n ≥ 5 for immunostainings; IKK2^E-KO *Ripk1*^D138N/D138N^ n = 12 [P7-P8] & n = 20 [P136-P410] for H&E and n ≥ 3 for immunostainings). Scale bars, 50 μm. **(B)** Microscopic quantification of epidermal thickness (Epi. th.) and inflamed skin area (Infl. area) as well as quantification of TUNEL & CC3 positive cells on the skin sections from 7- to 8-d-old mice with the indicated genotypes. **(C)** qRT-PCR analysis of the mRNA expression of the indicated cytokines and chemokines in RNA isolated from the epidermis of 4-d-old mice with the indicated genotypes (data for *Ikk2*^fl/fl^ and IKK2^E-KO mice are same as in Fig 1 and are shown for comparison). Each dot represents individual mice. *P ≤ 0.05; **P ≤ 0.01; ***P ≤ 0.005. Arrows indicate skin lesions. Source data are available for this figure.

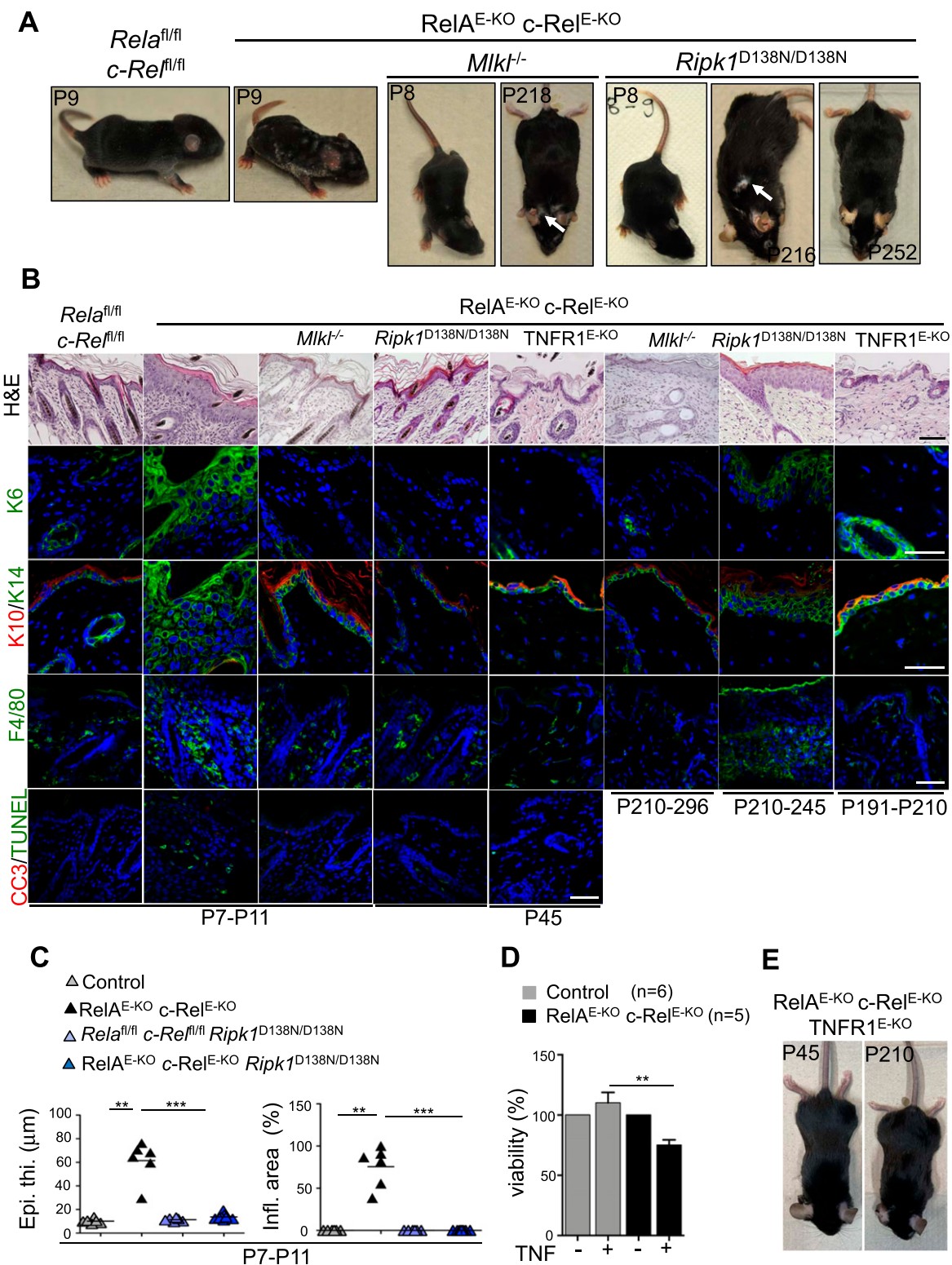

**Figure 6. NF-κB inhibition triggers RIPK1 kinase activity and mixed lineage kinase like-dependent necroptosis and skin inflammation.**
**(A, B)** Representative macroscopic pictures and skin sections from mice with the indicated age and genotype stained with H&E or immunostained with the indicated antibodies. Representative images are shown (RelA[E-KO] c-Rel[E-KO] n = 12 [P7-P10] for H&E and n ≥ 3 for immunostainings; RelA[E-KO] c-Rel[E-KO] Mlkl[−/−] n = 3 [P7-P11] & n = 6 [P130-P300] for H&E and n ≥ 3 for immunostainings; RelA[E-KO] c-Rel[E-KO] Ripk1[D138N/D138N] n = 11 [P7-P8] & n = 26 [P100-P364] for H&E and n ≥ 3 for immunostainings; RelA[E-KO] c-Rel[E-KO] TNFR1[E-KO] n = 3 [P45] & n = 4 [P161-P210] for H&E and n ≥ 3 for immunostainings). Scale bars, 50 μm. **(C)** Microscopic quantification of epidermal thickness (Epi. th.) and inflamed skin area (Infl. area) on the skin sections from 7- to 11-d-old mice with the indicated genotypes. Each dot represents individual mice. *P ≤ 0.05; **P ≤ 0.01; ***P ≤ 0.005. **(D)** Primary keratinocytes from Control (n = 6) or RelA[E-KO] c-Rel[E-KO] (n = 5) pups

RIPK1 kinase–dependent, necroptosis-mediated skin inflammation in RelA[E-KO] c-Rel[E-KO] was triggered by TNFR1, we generated RelA[E-KO] c-Rel[E-KO] mice lacking TNFR1 in keratinocytes. These triple deficient, RelA[E-KO] c-Rel[E-KO] TNFR1[E-KO], mice did not develop any signs of skin inflammation at least until the age of 6–7 mo, showing that TNFR1 expression in keratinocytes is essential for the development of inflammatory skin lesions in RelA[E-KO] c-Rel[E-KO] mice (Fig 6B and E). Taken together, these results showed that inhibition of NF-κB triggers skin inflammation by inducing TNFR1-RIPK1-MLKL–mediated keratinocyte necroptosis.

# Discussion

Numerous experimental and clinical studies have identified TNF as a potent cytokine that plays a crucial role in the pathogenesis of inflammatory diseases in human patients and in animal models (Chaudhari et al, 2001; Apostolaki et al, 2010; Peyrin-Biroulet, 2010; Wullaert et al, 2011; Monaco et al, 2015; Kumari & Pasparakis, 2017). However, the cellular and molecular mechanisms by which TNF causes chronic inflammatory conditions remain poorly understood. TNF binding to TNFR1 drives the expression of pro-inflammatory cytokines and chemokines primarily by activating NF-κB-dependent gene transcription. It was therefore counterintuitive that inhibition of IKK/NF-κB signaling in epidermal keratinocytes caused the spontaneous development of severe chronic skin inflammation in mice (Pasparakis et al, 2002; Kumari et al, 2013; Grinberg-Bleyer et al, 2015). Here we provide experimental evidence deciphering this paradox, by showing that IKK/NF-κB inhibition caused skin inflammation by triggering the TNF-mediated death of keratinocytes. Our genetic studies identified RIPK3-MLKL-dependent keratinocyte necroptosis as a key inducer of skin inflammation in IKK2[E-KO] and RelA[E-KO] c-Rel[E-KO] mice, consistent with the notion that necroptosis is an inflammatory type of cell death. In most experimental systems, inhibition of caspase-8 activity is required to sensitize cells to necroptosis (Pasparakis & Vandenabeele, 2015; Grootjans et al, 2017). It is therefore notable that keratinocytes lacking IKK2 or RelA and c-Rel underwent TNF-induced necroptosis in vivo even though they had intact FADD-caspase-8 signaling. Although it remains unclear how inhibition of IKK/NF-κB signaling sensitizes keratinocytes to necroptosis in the presence of intact caspase-8 activity, these results suggest that regulation of necroptosis within the tissue microenvironment may be different from the in vitro cell culture systems typically used for the study of necroptosis. This is further supported by studies showing that inhibition of necroptosis partially protects mice from systemic inflammatory response syndrome induced by injection of high dose recombinant TNF in the absence of caspase-8 inhibition (Duprez et al, 2011; Newton et al, 2016). Although blocking necroptosis by genetic ablation of MLKL or RIPK3 could strongly inhibit skin inflammation in IKK2[E-KO] and RelA[E-KO] c-Rel[E-KO] mice, these mice developed mild skin inflammation later in life indicating that another, necroptosis-independent, mechanism also contributes. Our findings that IKK2[E-KO] FADD[E-KO] Ripk3[−/−] mice were fully protected from skin lesion development provided experimental proof that FADD-caspase-8–dependent cell death is responsible for the mild skin inflammation observed in IKK2[E-KO] mice upon inhibition of necroptosis. Collectively, these findings unequivocally demonstrated that inhibition of IKK/NF-κB signaling in the epidermis causes skin inflammation by triggering keratinocyte death primarily by necroptosis, although FADD-caspase-8–mediated cell death could also contribute to skin inflammation in the absence of necroptosis.

Our genetic studies demonstrated that inhibition of RIPK1 kinase activity prevented keratinocyte death and skin inflammation in IKK2[E-KO] mice. This finding is consistent with the key role of IKK2 in preventing TNFR1-induced cell death by enforcing checkpoint 1 by directly phosphorylating RIPK1 preventing its activation and the induction of downstream cell death signaling (Dondelinger et al, 2015, 2019). Inhibition of RIPK1 kinase activity was also shown to prevent TNFR1-mediated skin inflammation in mice with Sharpin deficiency (Sharpin[cpdm/cpdm]) (Berger et al, 2014; Kumari et al, 2014; Laurien et al, 2020; Webster et al, 2020), which also disables checkpoint 1 by compromising linear ubiquitination of complex 1 signaling components. However, in contrast to the IKK2[E-KO] mice where skin inflammation was caused primarily by keratinocyte necroptosis, skin inflammation in Sharpin[cpdm/cpdm] mice was driven primarily by FADD-caspase-8–dependent keratinocytes apoptosis (Kumari et al, 2014; Rickard et al, 2014). Therefore, RIPK1 activation could cause skin inflammation by inducing different cell death signaling cascades downstream of TNFR1 in the two models, primarily RIPK3-MLKL–induced necroptosis in IKK2[E-KO] and mainly FADD-caspase-8–induced apoptosis in Sharpin[cpdm/cpdm] mice. Although the mechanisms determining the type of cell death induced by RIPK1 in response to IKK2 or Sharpin deficiency remain elusive at present, these findings provide in vivo experimental evidence that RIPK1-mediated cell death is a potent driver of skin inflammation.

Based on the currently accepted model of the two distinct checkpoints regulating TNFR1-induced cell death (Justus & Ting, 2015; Ting & Bertrand, 2016), we expected that keratinocyte death caused by combined ablation of RelA and c-Rel, which inhibits NF-κB–mediated gene transcription therefore disabling checkpoint 2, would not depend on RIPK1. In contrast to this prediction, our genetic studies demonstrated that inhibition of RIPK1 kinase activity prevented keratinocyte death and skin inflammation in RelA[E-KO] c-Rel[E-KO] mice. Consistent with these results, we and others showed previously that inhibition of RIPK1 kinase activity rescued the embryonic lethality of Rela[−/−] mice (Vlantis et al, 2016; Xu et al, 2018), providing evidence that NF-κB subunit deficiency caused RIPK1-mediated cell death not only in keratinocytes but also in the embryonic liver. Together, these results demonstrate that inhibition of NF-κB–dependent gene transcription caused RIPK1-mediated cell death in tissues of living mice and challenge the in vivo relevance of the two-checkpoint model of TNFR1-mediated cell death regulation, which is primarily based on in vitro studies in cultured cells. These results also imply that the transcriptional induction of NF-κB–dependent genes is critical to prevent RIPK1-dependent cell death. NF-κB regulates the expression of a large number of pro-survival factors, such as cIAP1/2, XIAP, BclxL, A20, and cFLIP (Oeckinghaus & Ghosh, 2009), which could be involved in preventing RIPK1-mediated cell death in keratinocytes. Further studies in relevant in vivo models will be required to dissect the role of these pro-survival factors and identify the key NF-κB–dependent mechanisms that are essential to prevent RIPK1 activation and the maintenance of health skin

were treated with TNF (20 ng/ml) for 18 h. Cell viability was determined by WST-1 assay. Graphs show mean ± SEM from pooled data from four independent experiments. **(E)** Representative macroscopic pictures of mice with the indicated age and genotype. Multiple comparisons of groups were evaluated by Kruskal–Wallis one-way ANOVAs with post-Dunn corrections. *$P \leq 0.05$; **$P \leq 0.01$; ***$P \leq 0.005$.

homeostasis. Whereas inhibition of RIPK1 kinase activity strongly prevented skin lesion development in both the IKK2$^{E-KO}$ and RelA$^{E-KO}$ c-Rel$^{E-KO}$ mice, some of these mice did develop very mild skin inflammation later in life suggesting that RIPK1-independent cell death also contributes, albeit to a much lesser extent. Taken together, our in vivo studies provided experimental evidence that TNFR1 drives severe skin inflammation in mice with keratinocyte-specific IKK/NF-κB inhibition by inducing RIPK1-mediated necroptosis and apoptosis, suggesting that RIPK1-dependent cell death could also contribute to human inflammatory skin diseases.

# Materials and Methods

$Ikk2^{fl/fl}$ (Pasparakis et al, 2002), $Fadd^{fl/fl}$ (Mc Guire et al, 2010), $K14$-$Cre$ (Hafner et al, 2004), $Mlkl^{-/-}$ (Dannappel et al, 2014), $Ripk1^{D138N/D138N}$ (Polykratis et al, 2014), $Rela^{fl/fl}$ (Luedde et al, 2008), $c$-$Rel^{fl/fl}$ (Heise et al, 2014), $Tnfr1^{fl/fl}$ (Van Hauwermeiren et al, 2013), $Ripk3^{-/-}$ (Newton et al, 2004), and $Ripk3^{fl/fl}$ (Newton et al, 2016) were described previously. Mice were kept at the specific pathogen-free animal facilities of the Institute for Genetics and the CECAD Research Center, University of Cologne under a 12 h light cycle, and given a regular chow diet. GF IKK2$^{E-KO}$ mice were generated and kept at the gnotobiotic facility of the Institute for Laboratory Animal Science and Central Animal facility (Hannover Medical School). Animals received appropriate care and were euthanized when they reached pre-determined criteria based on the development of macroscopically visible skin lesions to minimize suffering. Mice were assigned at random to groups within specific genotypes. Mouse studies were performed in a blinded fashion. All animal procedures were conducted in accordance with European, national and institutional guidelines and protocols and were approved by local governmental authorities (Landesamt für Natur, Umwelt und Verbraucherschutz Nordrhein-Westfalen, and the Lower Saxony State Office for Consumer Protection and Food Safety [LAVES]).

## Histological analysis and immunostainings of tissue sections

Skin from mice was embedded in paraffin or snap frozen in OCT compound. Antigen retrieval for paraffin sections was performed in citrate buffer, pH6. Keratin 14 (MS-115; Neomarkers), Keratin 6 (PRB-169P; Covance), Keratin 10 (PRB-159P; Covance), TUNEL (Promega)/ Active caspase-3 (9661; Cell Signalling) staining was performed on paraffin, and F4/80 (clone A3-1, homemade or MCA497G; Bio-Rad) staining on cryo sections. The staining was visualized with Alexa-488 or Alexa-567 conjugated secondary antibody. Measurement of epidermal thickness and inflamed area was done as described before (Jiao et al, 2020). Briefly, five optical fields per section were measured to quantify epidermal thickness. In each field, four measurements were performed. Percentage of inflamed area was determined as the percentage of inflamed versus total number of optical fields at 20× on individual skin section. Quantification of TUNEL and CC3 staining was performed on five optical fields per sections at 20× magnification. All images were acquired using either a Zeiss Meta 710 confocal, Leica SCN400 slide scanner or Leica DM5500 B microscope, and processed with ImageJ, Leica digital image hub or Leica Aperio ImageScope software.

## Keratinocytes isolation, culture, and cell death assessment

Keratinocytes from newborn pups were isolated as described using dispase II (D4693; Sigma-Aldrich). Briefly, the skin from newborn mice was incubated overnight at 4°C in dispase II. After incubation, epidermis was separated and placed in TrypLE (12605-010; Gibco) for 20 min followed by making single cell suspension using low Ca$^{2+}$ DMEM/Ham's F12 medium (F 9092-0.46; Biochrom) with 10% chelax-treated FCS and supplements. Cells were grown in the Ca$^{2+}$ DMEM/Ham's F12 medium. For cell death assay, 20–25,000 keratinocytes were seeded in collagen coated 96 well plates, stimulated with 20 ng/ml TNF with or without Z-VAD (20 $\mu$M) for 18 h and viability was assessed with WST-1 assay as per manufacturer's instruction (Roche).

## RNA isolation, cDNA synthesis, and quantitative RT-PCR

Total RNA from skin tissue was extracted with Trizol Reagent (Life Technologies) and purified on RNeasy Columns (QIAGEN). cDNA was prepared using Superscript III cDNA synthesis Kit (Life Technologies). qRT-PCR of $Il1$-$b$, $Il$-$6$, $Il$-$24$, $Ccl3$, and $Ccl4$ genes was performed with Sybr green. $Hprt$ was used as reference genes. Primer sequences were used as described before (Kumari et al, 2013). qRT-PCR of $Cflar$, $Birc2$, $Birc3$, and $Birc5$ was performed with Taqman using QIAGEN primers. Data were analyzed according to the ΔCT method.

## Separation of epidermis

Mouse epidermis was separated after incubating skin samples in 0.5 M ammonium thiocyanate (NH$_4$SCN) in phosphate buffer, pH 6.8 (0.1 M NH$_2$HPO$_4$ and 0.1 M KH$_2$PO$_4$) for 30 min on ice.

## Statistical analysis

Data shown in column graphs represent mean ± SEM. For statistical analysis of data from qRT-PCR, quantification of epidermal thickness and inflamed area, when data did not fulfill the criteria for Gaussian distribution, nonparametric Mann–Whitney test was performed. For the keratinocyte's death experiments, multiple comparisons of groups were evaluated by ordinary or Kruskal–Wallis one-way ANOVAs with post-Dunn corrections. *$P \leq 0.05$; **$P \leq 0.01$; ***$P \leq 0.005$. Statistical analysis was performed using GraphPad Prism.

# Supplementary Information

# Acknowledgements

We thank E Gareus, J Kuth, B Kühnel, E Stade, and C Uthoff-Hachenberg for excellent technical assistance. We thank V Dixit and K Newton, Genentech Inc, for $Ripk3^{-/-}$ and $Ripk3^{fl/fl}$ mice, and G Kollias for $Tnfr1^{fl/fl}$ mice. Research

reported in this publication was supported by funding from the Deutsche Forschungsgemeinschaft (DFG, German Research Foundation); projects SFB829 (project no. 73111208), SFB1403 (project no. 414786233), and under Germany's Excellence Strategy–EXC 2030 CECAD (project no. 390661388) to M Pasparakis, SPP1656 (project number 220134278) to M Pasparakis and A Bleich and BL953/5-2 to A Bleich.

## Author Contributions

S Kumari: conceptualization, data curation, formal analysis, supervision, validation, investigation, visualization, methodology, and writing—original draft, review, and editing.
T-M Van: formal analysis, validation, investigation, and methodology.
D Preukschat: formal analysis, validation, investigation, and methodology.
H Schuenke: formal analysis, validation, investigation, and methodology.
M Basic: investigation and methodology.
A Bleich: resources, investigation, and methodology.
U Klein: resources.
M Pasparakis: conceptualization, resources, formal analysis, supervision, funding acquisition, investigation, project administration, and writing—original draft, review, and editing.

## Conflict of Interest Statement

The authors declare that they have no conflict of interest.

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
