## [Reviewer comments · Life Science Alliance]

Life Science Alliance

NF- κ B inhibition in keratinocytes causes RIPK1-mediated necroptosis and skin inflammation

Manolis Pasparakis, Snehlata Kumari, Trieu My Van, Daniela Preukschat, Hannah Schuenke, Marijana Basic, André Bleich, and Ulf Klein

DOI: <https://doi.org/10.26508/lsa.202000956>

Corresponding author(s): Manolis Pasparakis, University of Cologne

Review Timeline:

Submission Date:	2020-11-13
Editorial Decision:	2020-12-14
Revision Received:	2021-03-15
Editorial Decision:	2021-03-26
Revision Received:	2021-04-01
Accepted:	2021-04-01

Scientific Editor: Shachi Bhatt

Transaction Report:

December 14, 2020

Re: Life Science Alliance manuscript #LSA-2020-00956-T

Prof. Manolis Pasparakis
University of Cologne
Institute for Genetics
CECAD Research Center
Joseph-Stelzmann-Str. 26
Cologne 50931
GERMANY

Dear Dr. Pasparakis,

Thank you for submitting your manuscript entitled "NF- κ B/IKK signaling prevents skin inflammation by inhibiting RIPK1 kinase activity-dependent necroptosis" to Life Science Alliance. The manuscript was assessed by expert reviewers, whose comments are appended to this letter.

As you will note from the reviewers' comments below, the reviewers were quite enthusiastic about this manuscript, but they have also suggested some additional experiments that will improve the message and robustness of this study. Thus, we encourage you to submit a revised manuscript to Life Science Alliance that addresses all of the reviewers' points.

Thank you for this interesting contribution to Life Science Alliance. We are looking forward to receiving your revised manuscript.

Sincerely,

Shachi Bhatt, Ph.D.
Executive Editor
Life Science Alliance
<https://www.lsjournal.org/>
Tweet @SciBhatt @LSAJournal

- A letter addressing the reviewers' comments point by point.
- An editable version of the final text (.DOC or .DOCX) is needed for copyediting (no PDFs).
- High-resolution figure, supplementary figure and video files uploaded as individual files: See our detailed guidelines for preparing your production-ready images, <https://www.life-science-alliance.org/authors>
- Summary blurb (enter in submission system): A short text summarizing in a single sentence the study (max. 200 characters including spaces). This text is used in conjunction with the titles of papers, hence should be informative and complementary to the title and running title. It should describe the context and significance of the findings for a general readership; it should be written in the present tense and refer to the work in the third person. Author names should not be mentioned.

B. MANUSCRIPT ORGANIZATION AND FORMATTING:

Reviewer #1 (Comments to the Authors (Required)):

Kumari et al described a study of NFkB/IKK signaling in suppressing RIPK1-mediated necroptosis in the skin. The power of genetics is well demonstrated in this study, providing compelling evidence that in the skin the pro-inflammatory signaling pathway mediated by Inhibitor of kappa B kinases (IKK) paradoxically suppresses inflammation induced by necroptosis through RIPK1/RIPK3/MLKL. Conditional deletion of IKK2 results in skin inflammation, which can be reversed by deletion of RIPK3 or MLKL. Similar, milder, skin inflammation also develops in mice conditionally lacking Rela and c-Rel

NFkB genes, which is ameliorated by deletion of RIPK3 or MLKL. This latter observation argues against a previous two-check point model by Ting et al. in TNFR1-induced death responses in cell lines. Lack of FADD provides further improvement, indicating IKK and NFkB restrict both necroptosis and apoptosis in the skin. The genetic data is compelling, and the study carries significant impact in the field. Some revision is recommended.

-The authors used cc3 as a marker of death in IHC assays. Activation of caspases is a hallmarks of apoptosis. Given the argument that in the absence of IKK2 or RelA/cRel, necroptosis is a major form of death, it is recommended that IHC be performed with antibodies specific to p-MLKL and pRIPK3, which are specific markers for necroptosis.

The Ripk1 D138N allele described by Polykratis et al., 2014 was also a floxed allele. If indeed so, the compound mice described in the current study is RIPK1 null in the skin, not D138N. Please clarify! If K14-Cre is a transgene, perhaps genetic designation should be K14-Cre+, not K14tg/wt. There is no heterozygosity for Tgs (only hemizyosity).

Reviewer #2 (Comments to the Authors (Required)):

In their manuscript, Kumari and colleagues investigate how defective NF-kB signaling affects skin inflammation. The authors conclude that the major outcome of deficient NF-kB signaling is RIPK1 activation. Consequently, inhibition of RIPK1 kinase activity blocks skin inflammation caused by IKK2 or Rel A/c-Rel deficiency. The study is very solid and comprehensive, and experiments are well controlled.

Major issues

One issue I have has to do with the title of this study. I would suggest that the authors consider the title that would more accurately reflect that "defective NF-kB signaling activates RIPK1 kinase activity dependent necroptosis".

The authors should also discuss more how their findings reflect on the possible NF-kB signaling components/proteins-mediated inhibitory RIPK1 phosphorylation and lack of it in the context of skin inflammation.

Specific points

- Line 92: TAK1/TAB2/3 complex is recruited to K63-linked ubiquitin chains, not to linear ubiquitin chains.
- Fig 1D: It would be good to show levels of TNF, IL-6 and IL-8 if possible.
- Line 227-229: Here and elsewhere when discussing Sharpin cpdm skin model it would be appropriate to reference Webster et al, JLB 2020 where it was shown that RIPK1 inhibition can block cpdm mediated skin apoptosis and necroptosis.
- Fig 3A and elsewhere: This is likely a difficult task but it is hard to see skin lesions in these images. Maybe point them in some way to make it easier for the reader.
- Line 242: If possible, please quantify cleaved caspase-3 and TUNEL positive cells. Cleaved caspase-3 quantification would be very helpful in several figures (3-6), especially in crosses with RIPK3 and MLKL knockouts where the absence of these critical necroptosis mediators nicely reduces skin inflammation.

Comments for the Editor

This manuscript can be accepted after the authors address listed comments. But overall, these are minor issues and I would invite the authors to resubmit revised version of their manuscript.

We would like to thank the reviewers for their insightful comments, which were helpful to further improve our manuscript. Below we include a detailed point by point response to all the comments.

In addition to all the changes we implemented in the manuscript in response to the reviewers' comments, we have also included additional data demonstrating the key role of TNFR1 in driving skin inflammation in *RelA^{E-KO} cRel^{E-KO}* mice. To further support that, similarly to *IKK2^{E-KO}*, necroptosis in *RelA^{E-KO} cRel^{E-KO}* mice is driven by epithelial TNFR1, we generated and analysed *RelA^{E-KO} cRel^{E-KO} TNFR1^{E-KO}* mice. These experiments showed that TNFR1 deficiency in keratinocytes prevents skin inflammation in *RelA^{E-KO} cRel^{E-KO}* mice, showing that similarly to IKK2 knockout, TNFR1 drives keratinocyte death and inflammation in response to epithelial NF- κ B inhibition. The new data is now included in Figure 6 of the revised manuscript.

Reviewer #1 (Comments to the Authors (Required)):

Kumari et al described a study of NF κ B/IKK signaling in suppressing RIPK1-mediated necroptosis in the skin. The power of genetics is well demonstrated in this study, providing compelling evidence that in the skin the pro-inflammatory signaling pathway mediated by Inhibitor of kappa B kinases (IKK) paradoxically suppresses inflammation induced by necroptosis through RIPK1/RIPK3/MLKL. Conditional deletion of IKK2 results in skin inflammation, which can be reversed by deletion of RIPK3 or MLKL. Similar, milder, skin inflammation also develops in mice conditionally lacking *Rela* and *c-Rel* NF κ B genes, which is ameliorated by deletion of RIPK3 or MLKL. This latter observation argues against a previous two-check point model by Ting et al. in TNFR1-induced death responses in cell lines. Lack of FADD provides further improvement, indicating IKK and NF κ B restrict both necroptosis and apoptosis in the skin. The genetic data is compelling, and the study carries significant impact in the field. Some revision is recommended.

-The authors used cc3 as a marker of death in IHC assays. Activation of caspases is a hallmarks of apoptosis. Given the argument that in the absence of IKK2 or *RelA/cRel*, necroptosis is a major form of death, it is recommended that IHC be performed with antibodies specific to p-MLKL and pRIPK3, which are specific markers for necroptosis.

We thank the reviewer for this valuable suggestion. We have indeed tried to detect the presence of necroptotic cells in the skin of our mice using immunostaining with antibodies against phosphorylated MLKL and RIPK3. Unfortunately, we have not been able to obtain specific and reproducible staining for p-MLKL or p-RIPK3 despite extensive efforts using all commercially available antibodies. This is not the case only in the *IKK2^{E-KO}* and *RelA^{E-KO} cRel^{E-KO}* mice, but in fact in all our mouse models for which we have obtained unequivocal genetic evidence that necroptosis drives the tissue pathology. Because of this problem, we have not been able to obtain direct evidence for the presence of necroptotic cells in the skin of our mice. As an alternative approach to obtain indirect evidence for necroptosis, we chose to perform double staining combining TUNEL assay with anti-cleaved caspase 3 (CC3) antibodies. Using this approach, we can differentiate between cells undergoing apoptosis, which are expected to be CC3+ or CC3/TUNEL double positive) and cells that undergo non-apoptotic cell death, which should be TUNEL+ but CC3-. As shown in Figure 2B, we detected a considerable number of TUNEL+CC3- cells in the epidermis of *IKK2^{E-KO}* mice, which provides evidence that IKK2-deficient keratinocytes undergo non-apoptotic cell death. Although this is not direct evidence of necroptosis, together with our genetic findings using *Ripk3^{-/-}* and *Mlkl^{-/-}* mice these results provide further support that necroptosis plays a critical role in driving the inflammatory skin disease in these mice.

The Ripk1 D138N allele described by Polykratis et al., 2014 was also a floxed allele. If indeed so, the compound mice described in the current study is RIPK1 null in the skin, not D138N. Please clarify!

This must be a misunderstanding, as the RIPK1D138N alleles in *Ripk1*^{D138N/D138N} mice are not floxed, as reported in our manuscript by Polykratis et al. The reviewer likely confused our mice with the RIPK1K45A mice reported by Berger et al in JI in 2014 (doi/10.4049/jimmunol.1400499), which indeed are also floxed.

If K14-Cre is a transgene, perhaps genetic designation should be K14-Cre+, not K14tg/wt. There is no heterozygosity for Tgs (only hemizyosity).

The designation of the genotype of transgenic mice is indeed an issue that often causes confusion. As the reviewer rightly points out, traditionally transgenes have been designated as 'positive' (e.g. *K14-Cre+*) or negative. We also used to apply this nomenclature in the past, however, with the introduction of colony management software we had to revisit the nomenclature of our mice including the transgenic lines in order to apply universal rules. Considering that every transgene represents a specific genetic locus (meaning the transgene insertion within a certain chromosomal location), it can also exist in homozygosity, although in the vast majority of cases transgenic mice are kept heterozygous. Therefore, transgenic loci can also be designated as tg or wt with all possibilities existing (tg/tg, tg/wt, wt/wt). As this nomenclature is more suitable to ensure a consistent genotype designation for the purposes of the colony management software, we have switched to using the tg/wt genotype for heterozygous transgene expression.

Reviewer #2 (Comments to the Authors (Required)):

In their manuscript, Kumari and colleagues investigate how defective NF-κB signaling affects skin inflammation. The authors conclude that the major outcome of deficient NF-κB signaling is RIPK1 activation. Consequently, inhibition of RIPK1 kinase activity blocks skin inflammation caused by IKK2 or Rel A/c-Rel deficiency. The study is very solid and comprehensive, and experiments are well controlled.

Major issues

One issue I have has to do with the title of this study. I would suggest that the authors consider the title that would more accurately reflect that "defective NF-κB signaling activates RIPK1 kinase activity dependent necroptosis".

We thank the reviewer for this suggestion. We have also been contemplating about the most informative title for our manuscript. We agree with the reviewer that a title stating that impaired NF-κB signaling activates RIPK1-dependent necroptosis better reflects the findings. We have therefore changed the title of the manuscript to: "NF-κB inhibition in keratinocytes causes RIPK1-mediated necroptosis and skin inflammation"

The authors should also discuss more how their findings reflect on the possible NF-κB signaling components/proteins-mediated inhibitory RIPK1 phosphorylation and lack of it in the context of skin inflammation.

We thank the reviewer for this suggestion. We agree that this is an important topic, which we refrained from discuss in more detail because at this stage we do not have data to support a specific mechanism. We have now included a brief discussion of the possible NF-κB-dependent factors that could be important in preventing RIPK1-dependent cell death in the discussion of our manuscript (lines 449-456).

Specific points

- Line 92: TAK1/TAB2/3 complex is recruited to K63-linked ubiquitin chains, not to linear ubiquitin chains.

We thank the reviewer for pointing out this wrong statement, which we have corrected in the revised manuscript.

- Fig 1D: It would be good to show levels of TNF, IL-6 and IL-8 if possible.

We have extensively assessed the expression of TNF in the skin of IKK2^{E-KO} mice in earlier work (Kumari et al, 2013) and found that TNF mRNA levels were not changed in the epidermis of these mice at P2 and P4. Increased TNF mRNA levels were detected in whole skin lysates at P6, indicating that transcriptional upregulation of TNF was a secondary consequence of the inflammatory response and not an early causative event. Our interpretation of this finding was that the amounts of TNF normally expressed in the skin are sufficient to drive the TNFR1-mediated signals that are essential for inducing keratinocyte necroptosis and inflammation. We have measured the expression of *Il-1b*, *Il-24*, *Ccl3*, *Ccl4* and *Il-6* and found that all these genes were upregulated in the epidermis of IKK2^{E-KO} mice at P4, but have not focused on homologues of IL-8. We have now included these results in new Fig. 1E.

- Line 227-229: Here and elsewhere when discussing Sharpin cpdm skin model it would be appropriate to reference Webster et al, JLB 2020 where it was shown that RIPK1 inhibition can block cpdm mediated skin apoptosis and necroptosis.

We thank the reviewer for pointing out the overlooked reference, which we have now cited in the revised manuscript, specifically in lines 140, 247 and 449.

- Fig 3A and elsewhere: This is likely a difficult task but it is hard to see skin lesions in these images. Maybe point them in some way to make it easier for the reader.

We thank the reviewer for this suggestion. We have now added arrows to indicate the skin lesions on the mouse pictures.

- Line 242: If possible, please quantify cleaved caspase-3 and TUNEL positive cells. Cleaved caspase-3 quantification would be very helpful in several figures (3-6), especially in crosses with RIPK3 and MLKL knockouts where the absence of these critical necroptosis mediators nicely reduces skin inflammation.

We thank the reviewer for this suggestion. We have now quantified TUNEL, CC3 and TUNEL/CC3 positive cells in all figures. These results are now included in the revised manuscript in Fig 2B, Fig 3B, Fig 4B and Fig 5B.

March 26, 2021

RE: Life Science Alliance Manuscript #LSA-2020-00956-TR

Prof. Manolis Pasparakis
University of Cologne
Institute for Genetics
CECAD Research Center
Joseph-Stelzmann-Str. 26
Cologne 50931
Germany

Dear Dr. Pasparakis,

Thank you for submitting your revised manuscript entitled "NF- κ B inhibition in keratinocytes causes RIPK1-mediated necroptosis and skin inflammation". We would be happy to publish your paper in Life Science Alliance pending final revisions necessary to meet our formatting guidelines.

Along with the points listed below, please also attend to the following,

- please revise the legend for figure 1 so that the panels are introduced in order
- please use the [10 author names, et al.] format in your references (i.e. limit the author names to the first 10)
- please move your main, supplementary, and table figures legends to the main manuscript text after the references section
- please add ORCID ID for secondary corresponding-they should have received instructions on how to do so
- please add a Summary Blurb/Alternate Abstract in our system
- the labeling of panels in Figures 3B, 4A, 5A and 6B is somewhat confusing as it appears as though there are blank panels in the figure. Could you please clarify and possibly also improve the labeling to avoid such a confusion?

A. FINAL FILES:

B. MANUSCRIPT ORGANIZATION AND FORMATTING:

Sincerely,

Shachi Bhatt, Ph.D.
Executive Editor
Life Science Alliance

<https://www.lsjournal.org/>

Interested in an editorial career? EMBO Solutions is hiring a Scientific Editor to join the international Life Science Alliance team. Find out more here -

https://www.embo.org/documents/jobs/Vacancy_Notice_Scientific_editor_LSA.pdf

Reviewer #2 (Comments to the Authors (Required)):

The authors have addressed all reviewers' comments and this manuscript should be accepted for publication now.

April 1, 2021

RE: Life Science Alliance Manuscript #LSA-2020-00956-TRR

Prof. Manolis Pasparakis
University of Cologne
Institute for Genetics
CECAD Research Center
Joseph-Stelzmann-Str. 26
Cologne 50931

Dear Dr. Pasparakis,

Thank you for submitting your Research Article entitled "NF- κ B inhibition in keratinocytes causes RIPK1-mediated necroptosis and skin inflammation". It is a pleasure to let you know that your manuscript is now accepted for publication in Life Science Alliance. Congratulations on this interesting work.

DISTRIBUTION OF MATERIALS:

Again, congratulations on a very nice paper. I hope you found the review process to be constructive and are pleased with how the manuscript was handled editorially. We look forward to future exciting submissions from your lab.

Sincerely,

Shachi Bhatt, Ph.D.

Executive Editor

Life Science Alliance

<https://www.lsjournal.org/>
